# rRNA transcription is integral to phase separation and maintenance of nucleolar structure

Soma Dash[1]☯, Maureen C. Lamb[1]☯, Jeffrey J. Lange[1], Mary C. McKinney[1], Dai Tsuchiya[1], Fengli Guo[1], Xia Zhao[1], Timothy J. Corbin[1], MaryEllen Kirkman[1], Kym Delventhal[1], Emma L. Moore[1], Sean McKinney[1], Rita Shiang[2], Paul A. Trainor[1,3]*

**1** Stowers Institute for Medical Research, Kansas City, Missouri, United States of America, **2** Department of Human and Molecular Genetics, Virginia Commonwealth University, Richmond, Virginia, United States of America, **3** Department of Anatomy and Cell Biology, University of Kansas Medical Center, Kansas City, Kansas, United States of America

☯ These authors contributed equally to this work.

* pat@stowers.org

**Data Availability Statement:** Original data underlying this manuscript can be accessed from the Stowers Original Data Repository at http://

## Abstract

Transcription of ribosomal RNA (rRNA) by RNA Polymerase (Pol) I in the nucleolus is necessary for ribosome biogenesis, which is intimately tied to cell growth and proliferation. Perturbation of ribosome biogenesis results in tissue specific disorders termed ribosomopathies in association with alterations in nucleolar structure. However, how rRNA transcription and ribosome biogenesis regulate nucleolar structure during normal development and in the pathogenesis of disease remains poorly understood. Here we show that homozygous null mutations in Pol I subunits required for rRNA transcription and ribosome biogenesis lead to preimplantation lethality. Moreover, we discovered that *Polr1a*⁻/⁻, *Polr1b*⁻/⁻, *Polr1c*⁻/⁻ and *Polr1d*⁻/⁻ mutants exhibit defects in the structure of their nucleoli, as evidenced by a decrease in number of nucleolar precursor bodies and a concomitant increase in nucleolar volume, which results in a single condensed nucleolus. Pharmacological inhibition of Pol I in preimplantation and midgestation embryos, as well as in hiPSCs, similarly results in a single condensed nucleolus or fragmented nucleoli. We find that when Pol I function and rRNA transcription is inhibited, the viscosity of the granular compartment of the nucleolus increases, which disrupts its phase separation properties, leading to a single condensed nucleolus. However, if a cell progresses through mitosis, the absence of rRNA transcription prevents reassembly of the nucleolus and manifests as fragmented nucleoli. Taken together, our data suggests that Pol I function and rRNA transcription are required for maintaining nucleolar structure and integrity during development and in the pathogenesis of disease.

www.stowers.org/research/publications/LIBPB-1752.

**Funding:** This work was funded by the Stowers Institute for Medical Research (1008 to P.A.T), American Association for Anatomy Post-Doctoral Fellowship (2021 to S.D.), K99 (DE030972) from the National Institute for Dental and Craniofacial Research (to S.D.), F31 (DE032256) from the National Institute for Dental and Craniofacial Research (to E.L.M.) and R01 (DE13172) from the National Institute for Dental and Craniofacial Research (to R.S.). The funders had no role in study design, data collection and analysis, decision to publish, or preparation of the manuscript.

**Competing interests:** The authors have declared that no competing interests exist

## Author summary

Ribosomal RNA (rRNA) is the catalytic component of a ribosome, which translates messenger RNA into protein in every cell. The process of transcribing rRNA takes place in a membraneless internuclear organelle, the nucleolus, which when disrupted leads to tissue-specific defects known as ribosomopathies in humans. How disruption of rRNA transcription affects nucleolar structure is poorly understood. We generated four different mouse mutants of subunits of the RNA Polymerase (Pol) I complex, which transcribes rRNA. Disrupting rRNA transcription results in pre-implantation embryo lethality. Prior to lethality, the mutant embryos exhibit reduced rRNA and one large, condensed nucleolus, compared to 2–5 nucleoli in control embryos. Pharmacologically inhibiting rRNA transcription in human induced pluripotent stem cells (hiPSCs) tagged with labels for different compartments of the nucleolus results in either a condensed nucleolus similar to the Pol I subunit mouse mutants or fragmented nucleoli. The fate of the nucleolus in a cell depends on the phase of cell cycle it is in prior to Pol I inhibition. Altogether, our data suggests that Pol I mediated rRNA transcription is vital for maintaining nucleolar structure and integrity during development and is disrupted in association with disease.

## Introduction

Ribosome biogenesis is a fundamental process required for protein synthesis in all cells, and therefore, is vital for all cell survival, growth and proliferation [1]. Ribosome biogenesis takes place in the nucleolus, which is a membrane-less organelle within the nucleus [2]. The nucleolus is tripartite in structure with each compartment having a specific function in the process of ribosome biogenesis [3–5]. The fibrillar center (FC) constitutes the innermost compartment of the nucleolus and forms around actively transcribed ribosomal DNA (rDNA) [6–10]. Here, RNA Polymerase (Pol) I in concert with many associated proteins such as UBF, Treacle and Nucleolin (NCL) transcribes rDNA into pre-ribosomal RNA, which is then transported to the surrounding dense fibrillar center (DFC) where it is cleaved and modified by several rRNA processing proteins, two of which are Fibrillarin (FBL) and Nopp140. Processed rRNA is then assembled with ribosomal proteins (RP) and other factors to form the small and large ribosome subunits. This takes place in the granular component (GC) which surrounds the FC and DFC, and is mediated by many proteins such as Nucleophosmin1 (NPM1) [11,12]. The different compartments of the nucleolus are maintained as distinct and defined regions through immiscible phase separation [4]. Phase separation relies on weak interactions between RNA and proteins with intrinsically disordered regions (IDRs) [13,14], such as FBL [15] and NPM1 [16], to concentrate particular proteins and RNAs to specific compartments of the nucleolus. Importantly, previous studies have demonstrated that changes in the phase separation properties of a nucleolar compartment can alter the morphology and function of the nucleolus [17].

In the nucleolus, Pol I is responsible for rRNA transcription, which is a critical rate limiting step in the ribosome biogenesis process [18,19]. Pol I consists of thirteen subunits, of which Polr1a and Polr1b form the catalytic core, while Polr1c and Polr1d form a clamp holding the catalytic core together [20]. Mutations in Pol I subunits cause developmental defects that disproportionally affect craniofacial development. More specifically, mutations in *POLR1A* cause Acrofacial Dysostosis Cincinnati type (AFDCin), whereas mutations in *POLR1B*, *POLR1C* and *POLR1D* and the Pol I associated protein, *TCOF1*/TREACLE lead to Treacher Collins Syndrome (TCS) [21–25]. We have previously defined the mechanisms underlying the pathogenesis of craniofacial defects in AFDCin and TCS in association with Pol I loss-of-function. In Pol

I subunit and associated factor mutants, rRNA transcription is impaired, which alters the stoichiometry between rRNAs and RPs such that excess RPs [26] and possibly 5S rRNA bind to Mdm2 [27]. This interaction prevents Mdm2 binding to and ubiquitinating p53 for proteasomal degradation. Consequently, p53 accumulates resulting in apoptosis of neural crest cells, the precursors of most of the craniofacial skeleton [26,28–31]. p53 activation is a hallmark of nucleolar stress, which is accompanied by changes in nucleolar structure (reviewed in [32,33]). However, whether nucleolar structure is affected by the genetic loss of Pol I function and rRNA transcription in association with ribosomopathy pathogenesis remains unknown.

Both oocytes and zygotes lack the classic tripartite nucleolar organization of somatic cells. However, at the 2-cell stage, when the zygotic genome is activated and rRNA transcription begins, nucleolar precursor bodies (NPB) appear which have a compact fibrillar structure. At embryonic day (E) 2.5, when mouse embryos have 8–16 cells, NPBs begin to form small FCs that are partially surrounded by a DFC. The mature tripartite structure of the nucleolus is observed in blastomeres from the E3.5 blastocyst stage onwards (reviewed in [34]). Considering that nucleologenesis is dependent upon active rRNA transcription, we hypothesized that nucleolar structure would be altered in the Pol I mutants and that this alteration would lead to nucleolar stress and abnormal development.

Here we show that homozygous null mutations in Pol I subunits required for rRNA transcription leads to preimplantation lethality. *Polr1a*[-/-], *Polr1b*[-/-], *Polr1c*[-/-] and *Polr1d*[-/-] mutants exhibit defects in the structure of their nucleoli, as evidenced by a decrease in number of NPBs and a concomitant increase in nucleolar volume, which results in a single condensed nucleolus. We observe that upon pharmacological inhibition of Pol I, preimplantation embryos exhibit two distinct phenotypes: a single condensed nucleolus similar to Pol I genetic mutants, or fragmented nucleoli like bodies. We also observe the same two phenotypes in human induced pluripotent stem cells (hiPSCs) when Pol I function and rRNA transcription is inhibited. We determined that the single condensed round nucleolus phenotype is likely caused by changes in the phase separation properties of NPM1 in the GC. Furthermore, the fragmented nucleolar phenotype is a result of the inability of the nucleolus to reform following mitosis due to the lack of Pol I activity. Overall, these results suggest that perturbation of Pol I function, rRNA transcription and protein distribution within the nucleolus can affect the phase separation of its components resulting in changes in nucleolar structure, all of which leads to embryo lethality or the pathogenesis of ribosomopathies.

## Results

### Loss of function of Pol I subunits results in nucleolar defects and preimplantation lethality

To investigate the function of Pol I subunits and rRNA transcription during mouse preimplantation development and in the regulation of nucleolus structure, we utilized our previously generated *Polr1a*[-/-], *Polr1c*[-/-] and *Polr1d*[-/-] null mouse mutants [26], and generated a new *Polr1b*[-/-] null mutant mouse, collectively referred to as Pol I mutants hereafter. While control embryos develop to the blastocyst stage, the Pol I mutant zygotes undergo four rounds of cell division following fertilization before arresting at the 16-cell stage (Fig 1A). Pol I mutant embryos fragment and appear as a ball of cells with no distinct boundaries between individual blastomeres (Fig 1B).

Since Pol I drives rRNA transcription during nucleolar assembly [6,7], we therefore hypothesized that NPB structure would be disrupted in Pol I mutants. To assess for changes in NPB structure, we immunostained the Pol I mutants with markers which label rRNA and the three major compartments of the nucleolus. We used the Y10b antibody to label 5.8S rRNA in the

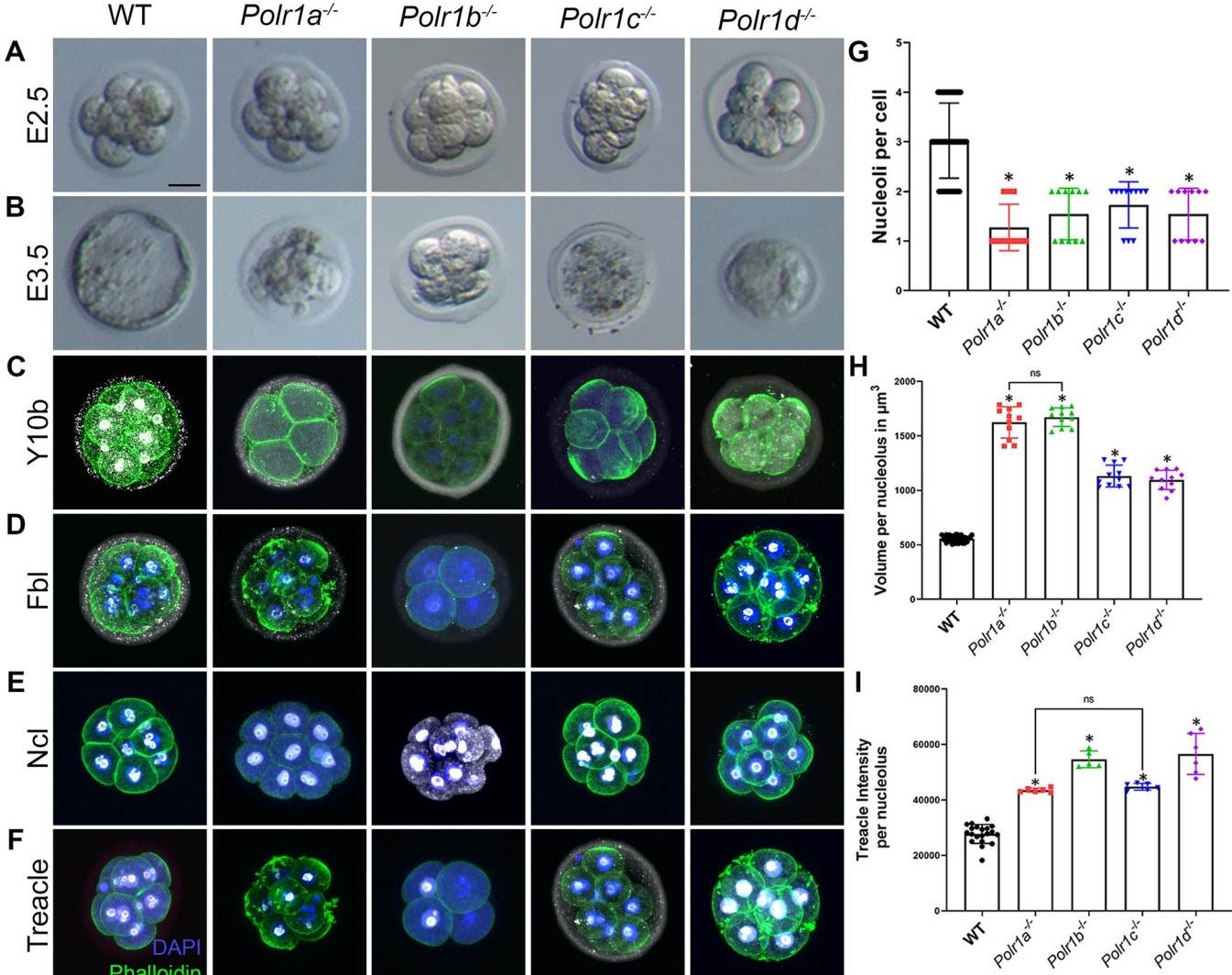

**Fig 1. Loss of function of Pol I subunits results in nucleolar defects.** Bright field images of WT, *Polr1a⁻/⁻*, *Polr1b⁻/⁻*, *Polr1c⁻/⁻* and *Polr1d⁻/⁻* embryos at E2.5 (A) indicate that the mutant embryos are indistinguishable from WT embryos at 8–16 cell stage. However, by E3.5 (B), Pol I mutant embryos are arrested in association with blastomere fragmentation while the WT embryos progress to the blastocyst stage. Immunostaining with Y10b to visualize rRNA (C) indicates that rRNA expression is reduced in all the mutant embryos. While Fbl (D) expression is variable, it is expressed in the NPBs of all the mutants. Ncl (E) and Treacle (F, I) expression are increased in the NPBs of all the mutants. In addition, the number of NPBs per blastomere is reduced in all the mutants (G), while the volume per NPB increases (H). The data is represented as mean+/-SEM. Scale bar for A-F = 12.5 μm. * indicates p<0.05, ns indicates p>0.05.

nucleolus as well as in the cytoplasm, and we observed a general downregulation of rRNA in Pol I mutant embryos (Fig 1C). This is consistent with our previous observations of decreased nascent rRNA transcription as measured by quantification of the 5' ETS of the 47S and 45S in Pol I subunit post-implantation mouse [26] and zebrafish embryos respectively [21,35,36]. We used Fbl, Treacle and Ncl expression to demarcate the FC, DFC and GC, respectively. We observe that the level of Fbl expression is variably affected within the different NPBs (Fig 1D), while Treacle and Ncl intensities are significantly and consistently increased within the NPB of Pol I mutants (Fig 1E, 1F and 1I). Furthermore, immunostaining for the FC, DFC and GC compartments revealed that the number of NPBs per cell decreased (Fig 1G) and the volume of NPB increased (Fig 1H) in Pol I mutants compared to wildtype embryos. These phenotypes are indicative of perturbed nucleolar structure in the absence of Pol I activity and reduced

rRNA transcription. In contrast, Pol I heterozygous null mutants do not exhibit nucleolar defects (S1 Fig).

## Loss of function of Pol I associated factor, Tcof1/Treacle, results in nucleolar defects and post-implantation lethality

Mutations in Pol I subunits are associated with AFDCin (*POLR1A*) and TCS (*POLR1B*, *POLR1C* and *POLR1D)* [37–39], ribosomopathies which are characterized by a constellation of craniofacial anomalies. Interestingly, however, the majority of individuals with TCS have a mutation in *TCOF1*, which encodes Treacle, a Pol I associated protein required for rRNA transcription [26,40]. We therefore hypothesized that *Tcof1* loss-of-function in mouse embryos would also result in embryo lethality in association with perturbed nucleolar structure [26]. Surprisingly, *Tcof1*$^{-/-}$ embryos survive until E10.5 similar to null mutations in other Pol I associated proteins such as Rrn3-TIF1A [41]. However, E7.5, *Tcof1*$^{-/-}$ embryos are smaller overall compared to wildtype littermates (S2A Fig), and by E8.5, *Tcof1*$^{-/-}$ embryos have a distinctly smaller head as well as delayed chorioallantoic fusion, which is indicative of defects in both ectoderm and mesoderm development (S2B Fig). Given the survival of *Tcof1*$^{-/-}$ embryos beyond gastrulation and the essential roles of Tcof1/Treacle in rRNA synthesis during development, [26,42,43], we hypothesized that Tcof1/Treacle may be maternally deposited and thus maternal protein activity continues in spite of the zygotic deletion of *Tcof1* in these embryos.

To test this hypothesis, we injected an antibody against Treacle into one blastomere of two-cell embryos to block the function of Treacle protein. The injected blastomere was also labelled with DiI to lineage trace its descendants. These mosaic embryos survive until the late blastocyst stage, but in contrast to control embryos fail to hatch (S2C and S2D Fig). We observed down-regulation of Treacle, 5.8S rRNA (as measured by Y10b expression), Fbl and Ncl in association with a single nucleolus in the Treacle antibody injected blastomeres, compared to non-injected blastomeres (S2E–S2L Fig). A similar downregulation of nucleolar proteins in association with fewer nucleoli was also observed in *Tcof1* mutant mouse embryonic fibroblast cells (MEFs) (S3 Fig). Altogether this data suggests that rRNA transcription is essential for nucleolar organization and structure during development.

## Nucleolar ultrastructure is altered in the *Polr1c*$^{-/-}$ mutant embryos

To better understand the effect of diminished Pol I function and rRNA transcription on nucleolar structure, we performed immunoelectron microscopy with Y10b and Ncl on *Polr1c*$^{-/-}$ mutants. We observed that the number of FCs and DFCs were significantly reduced in individual blastomeres (Fig 2A–A'). The staining patterns show that both rRNA and Ncl are redistributed to the edges of the GC, and that rRNA is diminished in the *Polr1c*$^{-/-}$ mutants (Fig 2B), consistent with the role of Pol I in rRNA transcription and ribosome biogenesis. Considering the FC and DFC of the NPBs are reduced in number and exhibit altered structure in *Polr1c*$^{-/-}$ mutants, we hypothesized this occurred in association with altered localization of active rDNA. Fluorescence in situ hybridization with probes designed to the 5'ETS of rDNA followed by immunofluorescence for Ncl revealed that rDNA loci were redistributed within the nucleus around the single nucleolus in *Polr1c*$^{-/-}$ mutants (Fig 2D–2D'), in a pattern reminiscent of nucleolar stress caps [44].

As described above, nucleolar disruption in Pol I mutants is accompanied by increased expression of Treacle in the NPB (Fig 1F and 1I). We hypothesized that in the absence of Pol I on the rDNA promoter and decreased rRNA transcription, perhaps Treacle remains bound to the rDNA promoter, or increasingly binds as a compensatory mechanism or as a means to prevent DNA damage [40]. The amount of biological material from preimplantation embryos is

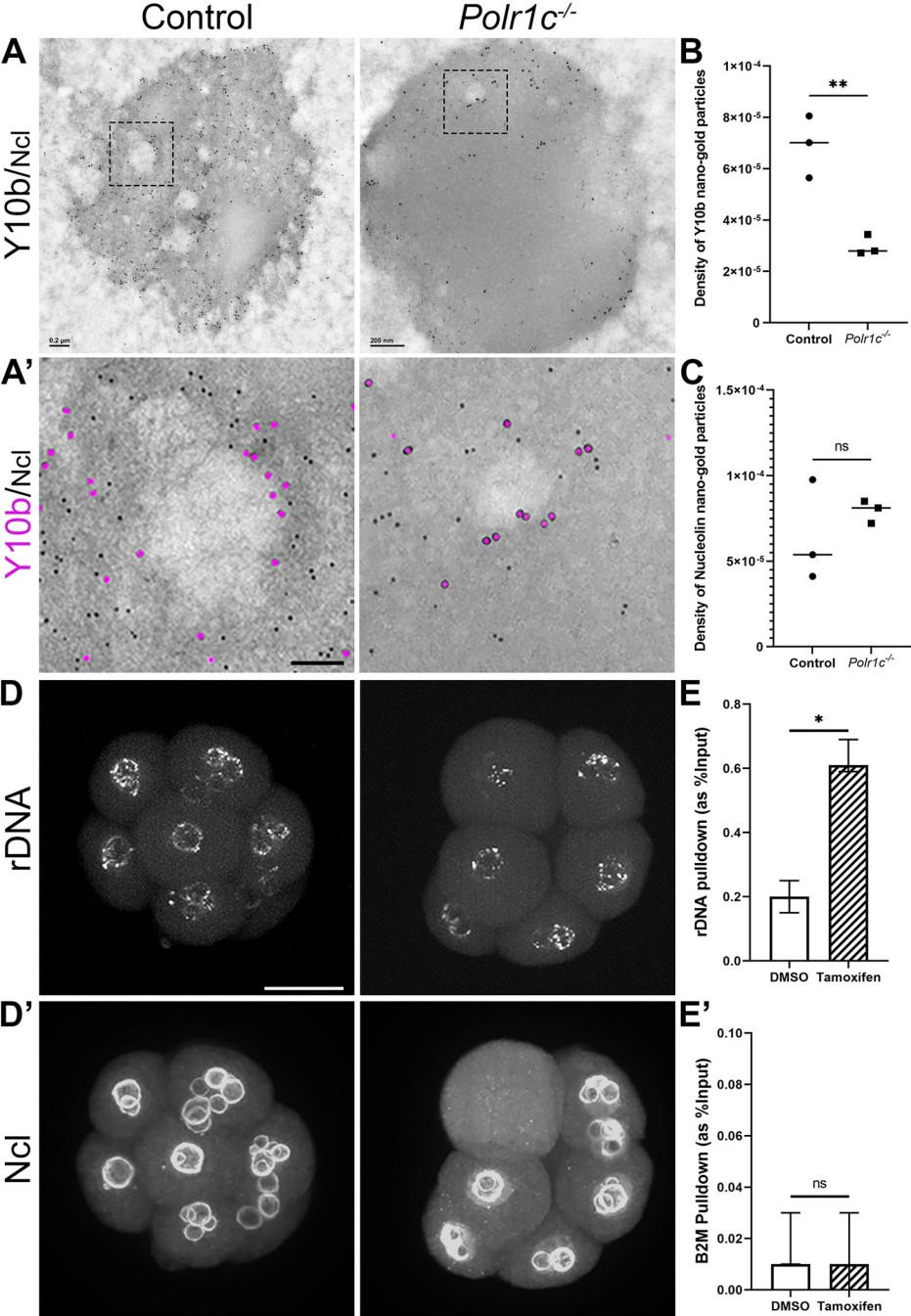

**Fig 2. Nucleolar organization is altered in the *Polr1c*<sup></sup> mutants.** Nucleolar organization is altered in the *Polr1c*^-/- mutants. (A) Immunoelectron microscopy with antibodies against Y10b (large particles) and Ncl (small particles) on *Polr1c*^-/- mutant embryos indicate that the number of FC and DFC are significantly reduced in *Polr1c*^-/- mutants compared to control embryos. Scale bar = 200 nm. (A') is higher magnification image of the FC/DFC shows lower levels of both Y10b (magenta) and Ncl (black) in the *Polr1c*^-/- mutants. Scale bar = 75 nm. Quantification (Mean+/-SEM) indicates that while Y10b is significantly reduced in *Polr1c*^-/- mutants (B), Ncl is not (C). 3D-immuno FISH with Ncl antibody (D'), and 5'ETS region of the 47S rDNA (D) indicates rDNA localization to the condensed nucleolus. Scale bar = 15 μm. (E) ChIP with Treacle antibody pulls down a significantly higher amount of rDNA promoter in tamoxifen treated *Polr1c*^fx/fx;*Cre-ER*^T2 MEFs compared to DMSO treated MEFs. B2M promoter was used as an internal control (E'). Data is represented as mean +/- upper and lower limits. * indicates $p<0.05$, ** indicates $p<0.01$, ns indicates $p>0.05$.

too low to perform a chromatin immunoprecipitation (ChIP) assay, therefore we used *Pol I* mutant MEFs to evaluate Treacle binding to rDNA promoters. First, we confirmed that the nucleolar defects in *Polr1a*$^{fx/fx}$*;Cre-ER*$^{T2}$ and *Polr1c*$^{fx/fx}$*;Cre-ER*$^{T2}$ MEFs, 48 hours after treatment with Tamoxifen, were similar to *Polr1a*$^{-/-}$ and *Polr1c*$^{-/-}$ embryos (S3 Fig). Next we performed ChIP with an antibody against Treacle with lysates prepared from *Polr1c*$^{fx/fx}$*;Cre-ER*$^{T2}$ MEFs treated with DMSO or Tamoxifen. qPCR after ChIP revealed a significantly higher amount of 47S rDNA promoter was pulled down with Treacle in *Polr1c* mutant MEFs compared to controls (see methods). Altogether, this suggests that in the absence of Pol I activity, Treacle either remains bound to the rDNA promoter, or its binding is increased as a compensatory mechanism or attempt to continue rRNA transcription. Either way, the absence of rDNA transcription results in disruption of nucleolar organization and function.

## Pharmacological inhibition of Pol I activity results in two distinct phenotypes

Pol I mutants exhibit a consistent reduction in the number of NPB that coalesce to a single large nucleolus. To unveil the mechanisms underlying this phenotype we treated preimplantation embryos with a pharmacological inhibitor of Pol I. We chose BMH-21 because it is a Pol I specific inhibitor that does not have off target effects, such as activation of the DNA damage response, which occurs with other Pol I inhibitors [45,46]. One- or two-cell mouse embryos cultured with BMH-21 or its solvent DMSO arrest at the 2-cell stage in association with blastomere fragmentation. Similarly, 4-cell stage embryos cultured with BMH-21 or DMSO fail to progress to the 8-cell stage. This fragmentation and perturbation of development is consistent with previous observations that DMSO inhibits the development of mouse embryos from 2-cells to 8-cells [47,48]. However, in contrast to control 8-cell embryos which develop to blastocysts, 8-cell embryos treated with BMH-21 undergo one round of cell division prior to embryo lethality. Furthermore, we observed two distinct phenotypes in these BMH-21 treated embryos: a single nucleolus (Fig 3B, yellow arrow) as was expected from our previous Pol I mutant data; and a fragmented nucleolus (Fig 3B, red arrow). The fragmented nucleolus manifests as speckles of Treacle, Ncl and Y10b throughout the nucleoplasm. Interestingly, E8.5 embryos treated with BMH-21 also present with a similar single nucleolus and fragmented nucleoli phenotypes (Fig 3C–3D"). The nucleolar fragments present in the BMH-21 preimplantation embryos are probably the result of persistent protein-protein interactions, independent of rRNA transcription and processing. Altogether, these data suggest that inhibition of Pol I function and rRNA transcription disrupts nucleolar organization and structure in association with perturbed embryo development.

## Pol I inhibition alters the phase separation properties of the nucleolus and its structure

A single nucleolus phenotype has previously been attributed to nucleolar stress [49], but how this occurs mechanistically remains poorly understood. We therefore investigated the dynamics and biophysical properties of nucleolar organization in real time through live imaging of hiPSC (Allen Institute), in which the endogenous locus of three proteins expressed in either the FC, DFC and GC, is tagged with a distinct fluorophore [50]. The fluorescent tags do not alter expression levels, patterns, or protein function in these cells [50]. Halo-tagged UBF labels the FC, GFP-tagged Fibrillarin (FBL) demarcates the DFC and an RFP-tagged Nucleophosmin1 (NPM1) marks the GC, allowing for real-time visualization of the dynamic morphology of each nucleolar compartment.

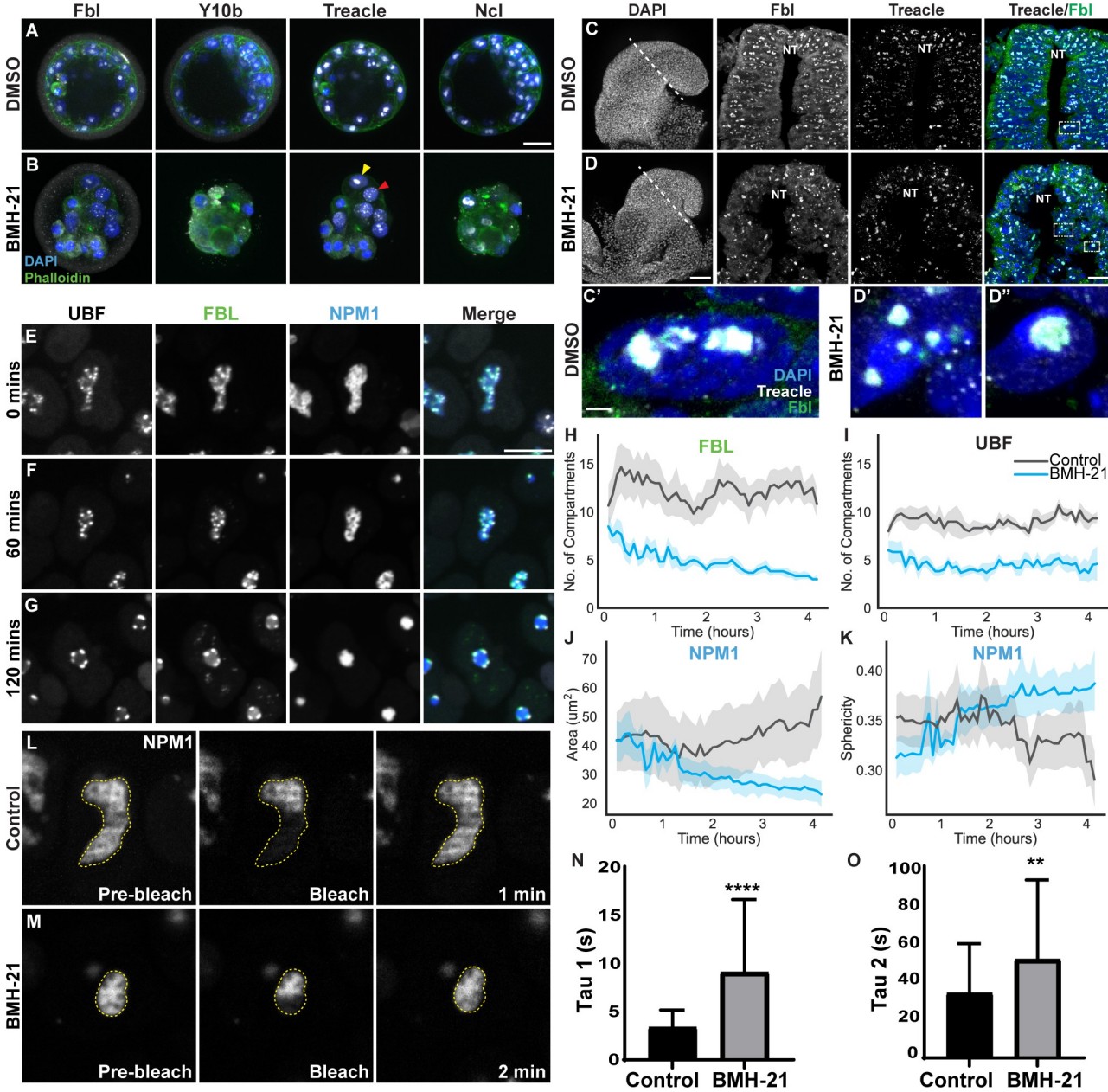

**Fig 3. Pol I inhibition alters nucleolar structure and phase separation properties.** (A, B) 16-cell stage DMSO and BMH-21 treated mouse embryos immunostained with Fbl, Y10b, Treacle and Ncl, and counterstained with DAPI and Phalloidin. The yellow arrow indicates a condensed nucleolus phenotype, while the red arrow points to a fragmented nucleolus phenotype. Scale bar = 20μm. (C, D) E8.0 embryos treated with DMSO and BMH-21 immunostained with Fbl and Treacle, and counterstained with DAPI. Scale bar = 50μm. (C'-D") High magnification images of cells from insets in C and D to show nucleolar disruption in BMH-21 treated embryos. Scale bar = 2.5μm (E-G) Stills from live imaging video of fluorescently tagged hiPSCs treated with BMH-21 showing the nucleolus condensing. Scale bar = 10μm. (H-K) Analysis of live imaging videos showing changes in the number of UBF (H) and FBL (I) puncta, NPM1 area (J), and NPM1 sphericity (K). n = 15 cells. Data is represented as mean (line) +/- 95% CI (shading). (L-M) Stills from half FRAP time-lapse video of pre-bleach, bleach, and recovery of NPM1 in control (L) and BMH-21 treated (M) cells. (N-O) Graphs of Tau values for half FRAP (N) and whole FRAP (O) showing both Tau1 and Tau2 significantly increase upon BMH-21 treatment (Student's t-test). The data is represented as mean+/-SD. ** indicates p<0.01, **** indicates p<0.0001.

First, we tested if inhibiting Pol I function and rRNA transcription using BMH-21 results in a nucleolar phenotype similar to that observed in Pol I mutant preimplantation embryos. Through live imaging of BMH-21 treated cells, we observed a reduced number of FCs (UBF,

grey) and DFCs (FBL, green) which unlike in Pol I mutant preimplantation embryos, formed characteristic nucleolar caps around the GC (NPM1, blue) that collectively coalesced into a single round nucleolus (Fig 3E–3I and S1 Video) [49]. In the GC, the NPM1 area decreases, and sphericity increases following BMH-21 treatment (Fig 3G, 3J and 3K), similar to the coalesced single nucleolus we observed in BMH-21 treated preimplantation embryos (Fig 3B, yellow arrow). These results indicate that when Pol I activity is perturbed, hiPSCs present with a similar nucleolar phenotype as BMH-21 treated and Pol I mutant preimplantation embryos.

Nucleolar structure and Pol I function are intimately linked in that perturbations in one affect the other. We hypothesized that the perturbation in nucleolar structure was caused by changes in the phase separation properties of the nucleolus upon loss of rRNA transcription. To test whether the phase separation properties of the nucleolus changed with Pol I inhibition, we performed Half Florescence Recovery After Photobleaching (FRAP) in hiPSCs. This method is frequently used to measure changes in liquid-like droplets *in vivo* [51,52]. We attempted to perform FRAP on FBL, however due to the size and shape of the DFC, performing half and whole FRAP caused phototoxicity to the hiPSCs. Therefore, we chose to analyze changes in phase separation using FRAP of NPM1 in the GC. We performed all FRAP experiments *in situ* in the hiPSC line to control for environmental or other *in vitro* factors that may contribute to the phase separation properties of the nucleolus.

For each FRAP experiment, we performed both whole and half FRAP. For whole FRAP the entire NPM1 area was instantaneously photobleached and recovery monitored. We hypothesized that whole FRAP recovery would only occur from nucleoplasmic NPM1 entering the nucleolus. Our results support this hypothesis. For the whole FRAP experiments (S4C–S4H Fig) we observed an exponential recovery in fluorescence that is comprised of a single diffusive component which we have labeled Tau. This Tau is the recovery time corresponding to the diffusion of NPM1 to the nucleolus. For each half FRAP experiment, half of the NPM1 area was instantaneously photobleached and monitored for fluorescence recovery (Fig 3L–3M and S2 Video). We hypothesized for the half FRAP experiments that if the NPM1 behaved in a liquid-like manner, we would observe a two-component recovery where NPM1 is added to the nucleolus from both the nucleoplasm and from the unbleached portion of the nucleolus. Supporting our hypothesis, we observed recovery that is comprised of these two different components, which we have labeled as Tau1 and Tau2, respectively. To properly account for the addition of nucleoplasmic NPM1 to the nucleolus in the half FRAP data, Tau2 in the half FRAP experiments was fixed to the Tau value from whole FRAP experiments. Tau1 for the half FRAP was allowed to fit to the data. A half FRAP Tau1 recovery that is faster than Tau2 would suggest liquid-like properties of either low (fast) or high (slow) viscosity in the nucleolus [51,52].

We found the half FRAP recovery (Tau1) of BMH-21 treated cells (average Tau1 = 9.05s) is significantly longer than for control cells (average Tau1 = 3.33s), which is indicative of slower NPM1 recovery in the GC. These results suggest the viscosity of the GC significantly increases with Pol I inhibition (Fig 3N). The whole FRAP recovery increased with BMH-21 treatment (Tau2, Fig 3O), suggesting there is reduced affinity of NPM1 for the GC or increased resistance of NPM1 addition to the GC. Another metric that we can extract from the data is the percent recovery. Here, we report the percentage of the initial fluorescence intensity that we observe at the end of fluorescence recovery. The whole FRAP percent recovery (S4B Fig) decreased slightly with BMH-21 treatment, indicating there is less nucleoplasmic NPM1 available to add to the nucleolus. During BMH-21 treatment, the GC of the nucleolus is significantly reduced in size (Fig 3J). However, the BMH-21 treated cells with the smaller GC have a slower recovery, indicating that the difference in recovery is independent of the size of the area photobleached therefore, and that we are likely underestimating the true increased viscosity of the nucleolus in the BMH-21 condition. Additionally, the intensity of the NPM1 in the GC is higher in the

BMH-21 treated vs the control cells. This is likely due to the decreased area of the GC following BMH-21 treatment which caused NPM1 to be more concentrated in the GC. However, since our FRAP experiments completely photobleached the NPM1 in all or half of the GC, the initial amount of NPM1 doesn't affect our results. Overall, our data suggests the single nucleolar phenotype is likely a result of the changes in the phase separation properties of the nucleolus, specifically the GC, which exhibits slower diffusion and higher viscosity, precipitated by the perturbation of Pol I function and decreased rRNA transcription.

## BMH-21 treatment prevents nucleolar reassembly following cell division

In addition to the coalesced single nucleolus phenotype, we also observe a fragmented nucleolar phenotype in a subset of cells in BMH-21 treated preimplantation (Fig 3B, red arrow), and midgestation embryos (Fig 3D) and hiPSCs (Fig 4L). Live imaging of control hiPSCs indicates that during mitosis the nucleolus disassembles leaving UBF puncta associated with rDNA (Fig 4A–4C). Following telophase, the nucleolus begins reassembly with DFC and GC components, such as FBL and NPM1, first aggregated into prenucleolar bodies (PNBs) in the nucleoplasm (Fig 4D). Once Pol I-mediated rRNA transcription restarts, the PNBs associate with the sites of Pol I transcription and the nucleolus reforms (Fig 4E–4F and S3 Video). In cells treated with BMH-21, the nucleolus disassembles similar to control cells (Fig 4G–4I), however following mitosis the nucleolus is unable to reassemble (Fig 4J–4L and S3 Video). This results in a fragmented nucleolar phenotype where individual UBF, FBL and NPM1 puncta disperse throughout the nucleoplasm (Fig 4L). Pol I inhibition with BMH-21 doesn't inhibit cell division, likely because the cells that are able to divide are beyond the final cell division G2/M checkpoint when BMH-21 is added [53]. To test if the fragmented phenotype is primarily observed in cells unable to reform their nucleolus following cell division when Pol I activity is inhibited, we synchronized cells in G2/M using VM-26 [54–56] and then released the cells with or without BMH-21 treatment. VM-26 pauses cells in late G2, but when removed allows their entry into M phase [54–56]. No significant cell death was observed from treatment with VM-26. Following synchronization, cells were immediately fixed or released into media, and we observed 0% of cells with a fragmented phenotype (Fig 4N, 4O and 4Q). BMH-21 treatment after synchronization with VM-26 resulted in a significant increase in the percentage of cells (14.4%) with fragmented nucleoli (Fig 4P–4Q) compared to the 7.5% of cells with fragmented nucleoli in unsynchronized BMH-21 treated cells (Fig 4M). These results suggest that inhibition of Pol I activity in cells able to undergo mitosis prevents the nucleolus from reassembling, resulting in a fragmented nucleolar phenotype.

Taken together, our data indicates that Pol I activity and rRNA transcription are important for nucleolar organization and structure in a cell cycle phase dependent manner. If cells are in G0/G1/S when Pol I function and rDNA transcription is inhibited, their nucleoli coalesce into a single large nucleolus. However, if cells are in G2/M at the time of inhibition, their nucleoli fragment, the difference being the inability to seed nucleolus reformation after mitosis in the absence of Pol I function and RNA transcription.

## Discussion

The nucleolus is the largest membrane-less organelle in eukaryotic cells and it has a well-established role in ribosome biogenesis and several cellular stress responses, including DNA damage and proteotoxic stress [57]. Nucleolar dysfunction has been linked to several human congenital disorders [1,21,24,58–61], cancer [62,63], neurodegenerative diseases [64,65], viral infections [66–68] and aging [69]. The assembly of the nucleolus is thought to be dependent on actively transcribing rDNA [70,71], however this had yet to be tested genetically. Pol I

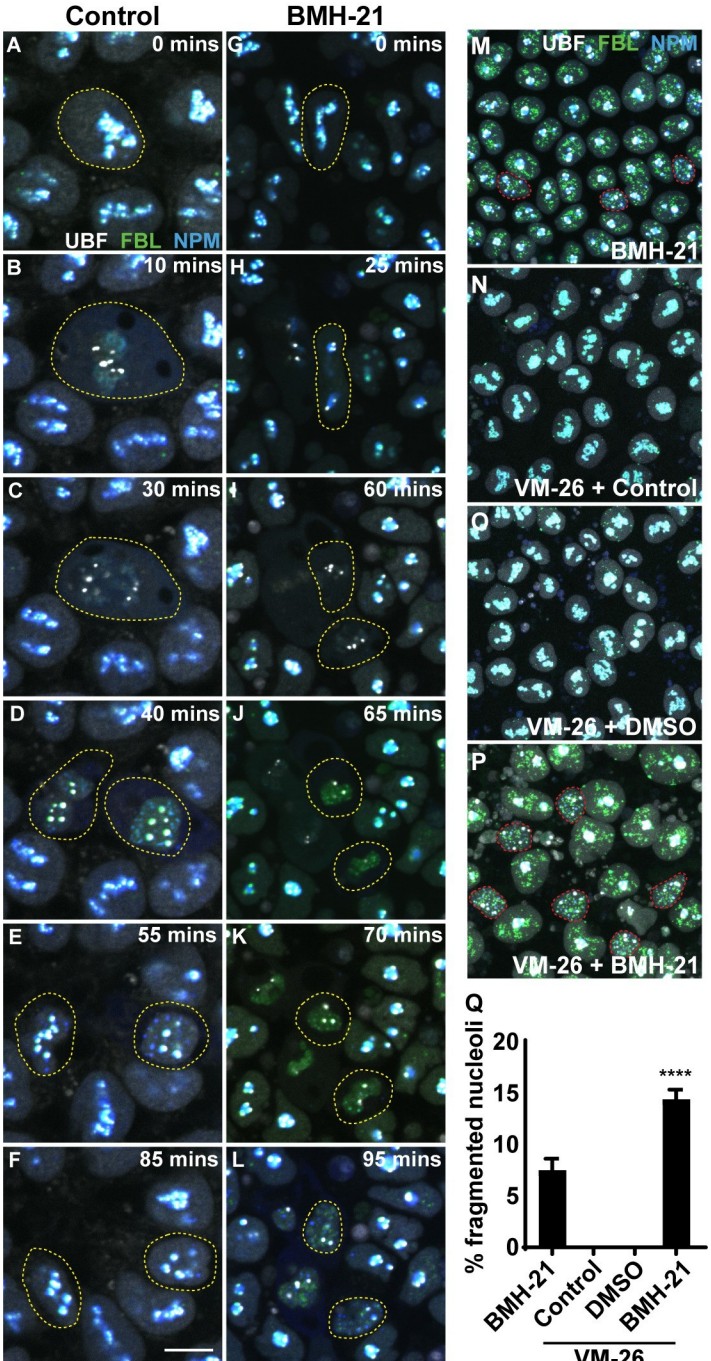

**Fig 4. BMH-21 inhibition of Pol I function prevents nucleolar reassembly following cell division.** (A-L) Stills from time-lapse imaging of control (A-F) and BMH-21 treated (G-L) hiPSCs progressing through the cell cycle. UBF = grey. FBL = green. NPM1 = blue. Yellow lines denote dividing cells. Scale bar = 10μm. (M-P) Confocal images of hiPSCs treated with BMH-21 (M), synchronized in G2/M with VM-26 (N), synchronized with VM-26 and released for 3 hours in regular media (O), and synchronized with VM-26 and released for 3 hours with BMH-21 treated media (P). Red lines denote cells with fragmented nucleoli. (Q) Graph of the percentage of cells with fragmented nucleoli indicating that BMH-21 treatment just prior to cell division increases percentage of cell with fragmented nucleoli. Data is represented as mean+/-SD. One-way ANOVA. **** indicates $p < 0.0001$.

mutant mouse embryos, in which rRNA is downregulated, survive until the 16-cell morula stage and present with a reduced number of NPBs and significantly increased volume of NPB. We hypothesize that these embryos survive until the 16-cell stage despite the loss of rRNA transcription due to maternal deposits of rRNA, ribosomes [30,72] and proteins, similar to what has been observed for the Subcortical Maternal Complex [73–76].

While the single nucleolus phenotype has previously been described as a consequence of perturbed ribosome biogenesis [77], the dynamic changes in structure and their underlying biophysical reasons remained to be characterized. We discovered that genetic deletion of Pol I subunits disrupts the transcription of rRNA, reducing the rRNA content and altering the localization or stability of nucleolar proteins such as Ncl and Treacle. Considering ribonucleoprotein particles maintain the phase separation properties of the nucleolus, we evaluated changes in the phase separation properties of the GC upon treatment with BMH-21 using FRAP. Our results indicate that perturbation of Pol I function with BMH-21 increases the viscosity of NPM1 in the GC. NPM1 typically exists as a pentamer [16] and contains both intrinsically disordered regions (IDRs) [78,79] that allow for homotypic interactions, and an RNA binding domain (RBD) which binds to the processed rRNA entering the GC [16,79,80]. Under normal conditions, NPM1 pentamers bind to rRNA through its RBD and assist in pre-ribosomal assembly until the ribosomal subunits are expelled from the GC [81]. However, when Pol I transcription is disrupted, which reduces the amount of rRNA entering the GC, this decreases NPM1-rRNA interactions and increases NPM1 homotypic interactions [81,82]. The increase in NPM1 homotypic interactions increases the density of the NPM1 meshwork and consequently GC viscosity [10,78,80], resulting in a single nucleolus. In fact, the single large nucleolus phenotype is consistent with the observation that the number of nuclear speckles, another membrane-less organelle in the nucleus, decreases upon transcriptional inhibition [83]. Furthermore, disruptions in Pol I transcription causes Polr1a to behave more like a liquid within nucleolar caps [84], again suggesting the activity of Pol I can influence the biophysical properties of the nucleolus. Therefore, inhibiting rDNA transcription leads to changes in NPM1 interactions in the GC and subsequently altered phase separation which perturbs nucleolar structure [82].

While nucleolar proteins, such as NPM1, can independently drive phase separation *in vitro* [10], rRNA also aids in maintaining phase separation of the nucleolus [85,86] and can alter phase separation droplet viscosity *in vitro* [85,87] by promoting the immiscibility of the DFC and GC through interactions with FBL and NPM1 [10,16,86]. Our data suggests that reducing rRNA transcription by genetically or chemically inhibiting Pol I limits NPM1 interactions with rRNA and other NPM1 molecules leading to redistribution of NPM1 and altered phase separation of the GC (S5 Fig). Although we were unable to perform FRAP of FBL in the DFC, we would also expect changes in the viscosity of FBL to occur as the concentration of rRNA decreases [85,88].

Interestingly, we also observe considerable relocalization of Ncl and Treacle in our Pol I mutant preimplantation embryos. Ncl contains a Gly-Arg-rich (GAR) domain (similar to FBL's IDR) and Treacle is also an intrinsically disordered protein [89–91]. Therefore, their change in localization could be due to changes in their interactions with rRNA or other proteins in the DFC, leading to changes in phase separation. Overall, these data demonstrate that impairing Pol I function and rRNA transcription leads to changes in the phase separation properties of the nucleolus, and this in turn alters nucleolar structure.

In hiPSCs, preimplantation and midgestation embryos treated with BMH-21, we observed two distinct phenotypes, a single condensed nucleolus, and fragmented nucleoli. The fragmented phenotype results from the inability of the nucleolus to reform following cell division. Typically, the nucleolus disassembles at the beginning of mitosis upon cessation of Pol I

transcription [92,93] and then reassembles following the re-initiation of transcription [94]. In early telophase, UBF, FBL, NCL and remaining pre-rRNA associate with active nucleolar organizer regions [70,71,95]. Also, during early telophase, DFC and GC factors, such as FBL and NPM1, are localized in discreet puncta in the nucleoplasm called pre-nucleolar bodies (PNBs) [70,71]. Upon Pol I activation in late telophase, the PNBs are recruited to sites of active rRNA transcription, and the fully formed nucleolus emerges. In BMH-21 treated hiPSCs, the nucleolus successfully disassembles preceding cell division, however, following cell division the nucleolus fails to reform and instead we observe discreet small puncta of UBF, FBL and NPM1 dispersed throughout the nucleoplasm. Our data demonstrates that when cells are synchronized in G2/M and Pol I activity is inhibited, an increased percentage of cells exhibit a fragmented nucleolar phenotype after division. This data suggests that the nucleolar phenotype, single condensed or fragmented, is dependent on the phase of the cell cycle when Pol I is inhibited.

The change in nucleolar structure in the absence of rRNA transcription most likely leads to nucleolar stress which is associated with p53 activation [26,28]. p53 activation in Pol I mutant and zebrafish embryos results in lethality [21,26,28,29,35] and in human patients leads to ribosomopathies. While DNA damage has been suggested to occur following perturbation of Pol I activity, we find no evidence for this *in vivo* in our genetic and other analyses [26]. Furthermore, BMH-21 has been shown to elicit its effect on Pol I function, independent of any DNA damage response [45].

Considering human patients affected by a ribosomopathy typically have a heterozygous mutation in a Pol I subunit and, or associated factor, it's important to consider whether nucleolar morphology and assembly are similarly impacted in cells where RNA Pol I activity is reduced but not completely lost. Recently, AFDCin disease associated POLR1A variants modeled in-vitro were shown to perturb rRNA synthesis in concert with altered nucleolar structure [96]. Therefore, changes in nucleolar structure, which result in nucleolar dysfunction, are part of ribosomopathy pathogenesis.

Overall, our work has uncovered the molecular and biophysical mechanisms underlying structural changes in the nucleolus resulting from genetic and chemical inhibition of Pol I function and rRNA transcription. Thus, Pol I function and nucleolar structure are intimately linked, in that perturbation in one affects the other during development and in the pathogenesis of disease.

## Methods

### Ethics statement

All animal experiments were conducted in accordance with the Stowers Institute for Medical Research Institutional Animal Care and Use Committee approved protocol (IACUC #2022–143) and the Virginia Commonwealth University Institutional Animal Care and Use Committee approved protocol #AM10025).

### Animals

The day a vaginal plug was observed in a time mated female was designated as embryonic day (E) 0.5. All mice were housed in a 16 hour light: 8 hour dark light cycle. *Polr1a*$^{+/-}$, *Polr1c*$^{+/-}$ and *Polr1d*$^{+/-}$ mice were generated and maintained as previously described [26]. To generate *Tcof1*$^{\Delta/\Delta}$ embryos, *Tcof1*$^{fx/fx}$ female mice [26] were crossed with Zp3-Cre mice to delete *Tcof1* in germline cells and then *Tcof1*$^{\Delta/+}$ males and females were intercrossed.

CRISPR-Cas9 technology was used to engineer a new mouse strain containing a deletion of exons 2 and 3 in the *Polr1b* gene. Potential guideRNA target sites were designed using CCTOP

[97], and then were evaluated using the predicted on-target efficiency score and off-target potential [98]. GuideRNA target sites were designed in intron 1 and intron 3 to fully delete exons 2 and 3. For both selected guideRNA targets, the sequence was ordered as Alt-R CRISPR-Cas9 crRNA from Integrated DNA Technologies (IDT). Each crRNA was hybridized with a universal tracrRNA (IDT) to form a full length guideRNA. Ribonucleoprotein (RNP) was prepared for microinjection with 10ng/µl of each full length guideRNA and 10ng/µl Cas9 protein (IDT, #1081059).

Tissue samples (ear and tail clips) from resulting animals were lysed using QuickExtract DNA Extraction Solution (Epicentre) followed by PCR at the specific genomic location. Amplification products were analyzed for a size shift using a LabChip GX (Perkin Elmer). Selected samples with deletion sized products were purified using ExcelaPure 96-Well UF PCR Purification Kit (Edge Bio) followed by Sanger sequencing.

## Brightfield imaging

Embryos were harvested at E2.5 by flushing the oviduct with 1ml M2 media as described previously [99]. The embryos were imaged using a Leica MZ16 microscope equipped with a Nikon DSRi1 camera and NIS Elements BR 3.2 imaging software.

## Immunostaining

The harvested embryos were transferred to a 4 well dish containing 4% PFA (Alfa Aesar via VWR Cat. No. AA43368-9M) for 10 minutes. Immunostaining was performed as described previously [100]. The following primary antibodies were used: Tcof1 (Abcam, #ab65212), Y10b (Abcam, # ab171119), Nucleolin (Abcam, #ab22758) and Fibrillarin (Abcam, # ab154806). The volume and intensity of staining was measured using IMARIS. All images for each stain were individually acquired with the same settings, and brightness and contrast were adjusted the same throughout.

## Mouse embryonic fibroblast cells

Mouse embryonic fibroblasts were derived from E13.5 $Polr1a^{fx/fx};Cre$-$ER^{T2}$, $Polr1c^{fx/fx};Cre$-$ER^{T2}$ and $Tcof1^{fx/fx};Cre$-$ER^{T2}$ embryos and cultured as described previously [26]. Control cells were treated with DMSO, while mutant cells were treated with 5µM Tamoxifen to recombine floxed alleles, and thereby delete the floxed gene. All experiments were performed 48 hours post tamoxifen treatment. All images for each stain were individually acquired with the same settings, and brightness and contrast were adjusted the same throughout.

## Immunoelectron microscopy

For immunogold labelling, embryos were fixed in 4% PFA, dehydrated in ethanol, and embedded in LR-White resin. Ultrathin sections of about 80nm in thickness were mounted on Formvar and carbon coated copper grids, then washed three times with PBS and three times with PBS containing 1% bovine serum albumin and 0.15% glycine, followed by 30 minutes blocking with 5% normal goat serum. Samples were incubated for 1 hour with the Ncl and Y10b primary antibodies at room temperature. After washing in PBS, samples were incubated for 1 hour with gold-conjugated secondary antibodies (Jackson ImmunoResearch, #115-205-166 and 111-195-144). Sections were stained with 2% uranyl acetate and observed under a FEI electron microscope at 80 kV. The specificity of the immunoreaction was assessed in all cases by omitting the primary antibodies from the labelling protocol and incubating the sections only in the gold-conjugated secondary antibodies.

Quantification was performed as follows. To localize individual gold particles, Laplacian of Gaussian filters of an appropriate size, depending on the image resolution and particle size, were applied and maxima found using plugins in Fiji [101]. First the larger 12nm particles were found, and then regions around them were eliminated to prevent them from being found when searching a second time for 6nm particles. These were then masked against a manually annotated nucleolar outline and the particles/nm$^2$ were computed for each image and particle size.

## 3D-immuno FISH

Embryos were fixed with 4% paraformaldehyde (PFA) for 10 minutes, rinsed and permeabilized with 0.5% TritonX-100/PBS for 60 minutes. RNA was digested with 200μg/ml of RNAse in PBS for 2 hours at 37˚C, and embryos were rinsed with PBS two times. Blocking was performed using superblock (Thermo scientific, # 37580) for 2 hours at room temperature, and embryos were incubated with anti-Nucleolin antibody (Abcam, #ab22758) for 20 hours at 4˚C. After three rinses with PBS, secondary antibody (Donkey Alexa 647 anti-Rabbit) (Invitrogen, #A-31573) was applied for 2 hours at room temperature and the embryos were washed twice with PBS. Post fixation was performed with 4% PFA for 10 minutes, followed by two washes with PBS, and the embryos were then dehydrated with ice cold 100% Methanol for 30 minutes. With a small volume of 100% Methanol, the embryos were deposited on microscope slides, which were air dried for 1 hour. Probe DNA (mouse rDNA BAC clone RP23-225M6, Empire genomics) was mixed with Hybridization buffer (50% formamide, 2X SSC, 1% dextran sulfate, 100ug/ml salmon sperm DNA), applied on the slides, and both genomic DNA and probe DNA were denatured simultaneously on a heating block at 80˚C for 8 minutes. Hybridization was performed at 37˚C for 16 to 20 hours. Slides were washed with 2X SSC for 5 minutes, 50% formamide/2X SSC for 15 minutes at 37˚C, 2X SSC for 10 minutes two times, and 2X SSC/ 0.1% Triton X for 10 minutes. Embryos were stained with 10 ug/ml of DAPI for 30 minutes at room temperature. Slides were mounted with ProLong gold antifade mountant (Invitrogen, # 36930). The embryos were imaged using Nikon Ti2 with CSU-W1 Spinning Disk. All images for each stain were individually acquired with the same settings, and brightness and contrast were adjusted the same throughout.

## Chromatin immunoprecipitation

To pulldown the rDNA promoter, chromatin immunoprecipitation was performed as described previously [102] using a Treacle antibody. Student's t-test was used for statistical analysis. The following primers used: rDNA_F: 5'–ATAAATGAAGAAAATAACTAA–3'; rDNA_R: 5'–TCTGGTACCTTCTTAATCACAGAT3'; B2M_F: 5'–CTTCTCTACTGGGTC CACCG–3'; B2M_R: 5'–CTGCTTATCGGCTCGGAAGA–3'.

## BMH-21 treatment on preimplantation embryos

8-cell embryos from CD1 pregnant dam were harvested using M2 media and cultured in KSOM media in 5% $CO_2$ at 37˚C. After 30 minutes of normalization in the culture conditions, the embryos were treated with 0.1 μM BMH-21 for 8 hours. The controls were treated with DMSO. After 8 hours, the culture media was changed to fresh KSOM media and the embryos were cultured for an additional 16 hours, after which they were fixed and immunostained.

## BMH-21 treatment on E8.5 embryos

E8.5 embryos from C57Bl/6 pregnant dam were harvested and cultured in 50% rat serum in DMEM-F12 media in 20% $CO_2$ at 37C. After 30 minutes of normalization in the culture

conditions, the embryos were treated with 1 μM BMH-21 for 8 hours. The controls were treated with DMSO. After 8 hours, the embryos were then fixed and immunostained. Further, the embryos were cryo-sectioned and imaged using Nikon Ti2 microscope with CSU-W1 Spinning Disk. All images for each stain were individually acquired with the same settings, and brightness and contrast were adjusted the same throughout.

### Antibody microinjections in 2-cell embryos

Immature C57BL/6J female mice (3–4 weeks of age) were utilized as embryo donors. The C57BL/6J females were superovulated following standard procedures with 5 IU PMSG (Genway Biotech, #GWB-2AE30A) followed 46 hours later with 5 IU hCG (Sigma, #CG5) and subsequently mated to fertile C57BL/6J stud males. Females were checked for the presence of a copulatory plug the following morning as an indication of successful mating. One-cell fertilized embryos at were collected from the oviducts of successfully mated females at 0.5dpc and placed in KSOM media in a $CO_2$ incubator at 37˚C, 5% CO2. Fertilized oocytes were cultured overnight to the two-cell stage for microinjection the following morning. Microinjection was performed using a Nikon Eclipse Ti inverted microscope equipped with Eppendorf Transfer-Man micromanipulators, Eppendorf CellTram Air for holding of embryos, and Eppendorf FemtoJet auto-injector. A small drop of M2 media was placed on a siliconized depression slide and approximately 20–30 C57BL/6J oocytes were transferred to the slide for microinjection. The slide was placed on the stage of the microscope and 2-cell embryos were injected at 200x. 1–2 pico liters total of 1mg/ml Treacle antibody was injected into one blastomere of the developing two-cell embryo using previously described techniques [99]. The control embryos were injected with water. DiI (ThermoFisher, #C7000) was added to the injection mix in both control and antibody injected embryos as a lineage tracer. Immediately following microinjection, the embryos were returned to the CO2 incubator in KSOM culture media and observed daily for embryo development. Blastocyst stage embryos were fixed and imaged for DiI, which was subsequently bleached followed by immunostaining of the embryos with Treacle, Y10b, Fbl and Ncl.

### Cell culture practices

The human induced pluripotent stem cell line used in this study, AICS-0086 cl.147, was generated as described previously [103,104]. This cell line can be obtained through the Allen Cell Collection (www.allencell.org/cell-catalog). Undifferentiated hiPSCs were maintained and passaged on plates coated with hESC-qualified Matrigel (Corning #354277) in mTeSR1 (Stem Cell Technologies #85850) supplemented with 1% penicillin/streptomycin (ThermoFisher, #15070063). Cells were passaged approximately every 3–5 days (70–85% confluency) using Accutase (Gibco, #11105–01) to detach cells. Cells were plated in mTeSR1 + 1% P/S and 10 μM Rock Inhibitor (Y-27632, StemCell Technologies, #72308). A detailed protocol for cell line maintenance can be found at www.allencell.org/sops SOP: WTC culture v1.7.pdf. For BMH-21 treatment, 1um BMH-21 (Sigma, #SML1183) was added to mTeSR1 media for either 1 hour for fixed imaging or after the first acquired frame for live imaging. For VM-26 treatment, 80nM of VM-26 (Sigma, #0609) was added to media for 24 hours to synchronize cells in G2/M before being rinsed off.

### Live imaging acquisition

HiPSCs (AICS-0086 cl.147) with RFP-tagged Nucleophosmin, GFP-tagged Fibrillarin and Halo-tagged upstream binding factor were plated on Matrigel coated Ibidi 35 mm μ-Dishes (Ibidi, # 81156). Before imaging, cells were treated with 200 nM of Janelia Fluor HaloTag

Ligand 646 for 30 minutes. The cells were then imaged using a CSU-W1 spinning disc (Yoko-gawa) coupled to a Ti2 microscope (Nikon) through a 60x Plan Apochromat objective (NA 1.45). Excitation of GFP occurred at 488 nm, RFP occurred at 561 nm, and Halo JF 646 occurred at 640 nm. The emissions were collected for 20 to 200 ms per frame through a standard filter onto a Flash 4 camera (Hamamatsu). Cells were imaged every 10 minutes for 5 hours. Great care was taken to reduce illumination as much as possible to avoid phototoxicity.

### Live imaging analysis

Individual cells were cropped from time-lapse videos and imported into Imaris (Bitplane, Inc.). A total of 12 untreated cells from 3 different experiments and 12 drug treated cells from 3 different experiments were analyzed. Surfaces were created for each component of the nucleolus using fluorescent labels and tracked over time. The number of FBL and UBF components, and the area and sphericity of the NPM1 component were averaged and plotted with the 95% confidence interval per time point for each treatment.

### FRAP acquisition and analysis

HiPSCs (AICS- 0086 cl.147) with RFP-tagged Nucleophosmin, GFP-tagged Fibrillarin and Halo-tagged upstream binding factor were plated on Matrigel coated Ibidi 35 mm μ-Dishes (Ibidi, #81156). The cells were then imaged using a CSU-W1 spinning disc (Yokogawa) coupled to a Ti2 microscope (Nikon) through a 60x Plan Apochromat objective (NA 1.45). Excitation of RFP occurred at 561 nm, and the emission was collected for 50 to 200 ms per frame through a standard filter onto a Flash 4 camera (Hamamatsu). Photobleaching was achieved using a diffraction-limited 561 nm laser beam focused on the region of interest and scanned across that ROI until fluorescence was eliminated. The settings for bleaching were adjusted so that bleaching was nearly instantaneous for both half and whole FRAP ($< 2$ s). Multiple prebleach images of cells were acquired for 0.5 s without delay, and the recovery after bleaching was recorded every 0.5s for 2 minutes. In separate instances, either the entire punctum was bleached (full FRAP) or only a part was bleached (half FRAP). Three independent experiments were performed for both control and BMH-21 treated cells with approximately 30 cells bleached for both half and whole FRAP in each experiment. Recovery curves and analysis were performed using in-house written plugins in Fiji (https://imagej.net/Fiji). First, the images were cropped to the cell of interest and then registered to remove the cell/punctum movement using a plugin called Stackregj. After this, an ROI was placed over the bleached portion of the cell and the mean intensity of the ROI was plotted using a plugin called "create spectrum jru v1." Once all the curves for a particular condition were collected, they were combined into one window using "combine all trajectories jru v1." The curves were then all normalized to the min and max of each curve using "normalize trajectories jru v1." The curves were then manually aligned in time so that the bleach points all aligned at the same timepoint. Finally, each curve was fit individually using "batch FRAP fit jru v1." The fit parameters were then averaged to give the Tau and percent recovery. The recovery for the whole FRAP experiments was fit by a single exponential yielding a single Tau value which corresponds to the addition of NPM1 to the nucleolus. Half FRAP data was fit by a double exponential yielding two Tau values. To properly account for the addition of protein to the aggregate in the half FRAP data, Tau2 in the half FRAP experiments was fixed to the Tau value from whole FRAP experiments. Tau1 for the half FRAP was allowed to fit to the data.

## Supporting information

**S1 Fig. Pol I heterozygous mutants have unaffected PNBs.** Immunostaining with Ncl to visualize NPBs indicates that its expression levels are consistent, and the number of NPBs per

blastomere are similar between WT and Pol I heterozygous mutants. Scale bar = 12.5 μm.
(TIF)

**S2 Fig. *Tcof1*<sup>-/-</sup> embryos are midgestation lethal. (A)** *Tcof1*<sup>-/-</sup> embryos are noticeably smaller than controls at E7.5. At E8.5 (B), these embryos are significantly smaller than their WT littermates. (C) Treacle antibody injected embryos exhibit cell death in the inner cell mass and fail to hatch, while control embryos proceed to hatching at E4.5 (D) Survival statistics of embryos injected with antibody indicates that over 50% of the embryos injected with Treacle antibody did not survive. Expression levels of Treacle (E, F), Y10b (G, H), Fbl (I, J) and Ncl (K, L) are significantly reduced in Treacle injected blastomeres and their descendants. The data is represented as mean+/-SEM. Scale bar for A and B is 100 μm. Scale bar for E, G, I and K = 12.5 μm. **** indicates p<0.0001.
(TIF)

**S3 Fig. Mutations in Pol I subunits causes Nucleolar Morphology defects in MEFs.** Immunostaining of MEFs with Treacle (A), Fbl (B), Ncl (C) and UBF (D) antibodies suggests that the number of nucleoli are significantly reduced in *Polr1a*<sup>fx/fx</sup>;*Cre-ER*<sup>T2</sup>, *Polr1c*<sup>fx/fx</sup>;*Cre-ER*<sup>T2</sup> and *Tcof1*<sup>fx/fx</sup>;*Cre-ER*<sup>T2</sup>. *Tcof1*<sup>fx/fx</sup>;*Cre-ER*<sup>T2</sup> MEFs have reduced expression of Fbl, Ncl and Ubf unlike *Polr1a*<sup>fx/fx</sup>;*Cre-ER*<sup>T2</sup> and *Polr1c*<sup>fx/fx</sup>;*Cre-ER*<sup>T2</sup>, compared to controls.
(TIF)

**S4 Fig. BMH-21 treatment causes increased recovery in whole and half FRAP GC. (A, B)** Graphs showing the percent recovery of the GC of half and whole FRAP respectively. Data is represented as mean+/-SD. Student's t-test. (C-H) Confocal images of whole FRAP NPM1 and its recovery over time in control and BMH-21 treated cells. (I-L) Average FRAP recovery curves for control (I, K) or BMH-21 treated (J, L) cells. The x-axis indicates slices or frames of the video where each slice or frame is the image taken at 0.5s intervals and the y-axis displays the intensity of the recovering NPM1 in the GC. ** indicates p<0.01, ns indicates p>0.05.
(TIF)

**S5 Fig. Nucleolar disruption in Pol I mutants. (A)** In control embryos and cells, each nucleolus has distinct FC and DFC regions surrounded by the GC and maintains an amorphous structure. rRNA is bound to nucleolar proteins (NPM1 in red, Ncl in purple and Treacle in blue). (B) When Pol I activity is inhibited, the nucleolus changes shape, becoming round and condensed. rRNA transcripts are reduced and nucleolar protein expression increases, which leads to a change in phase separation of the nucleolus. This leads to nucleolar stress in the pathogenesis of ribosomopathies.
(TIF)

**S1 Video. BMH-21 treatment of nucleolar tagged hiPSCs.** hiPSCs were imaged for UBF (HaloTag grey), FBL (GFP, green) and NPM1 (RFP, blue) every 10 minutes for 5 hours. 1μM of BMH-21 was added after the first frame was acquired. NPM1 condenses into a round circle while UBF and FBL form nucleolar caps at the GC periphery. Scale bar = 10μm.
(MP4)

**S2 Video. Half FRAP of control and BMH-21 treated hiPSC.** NPM1 (RFP, grey) was imaged in control (left) and BMH-21 treated (right) hiPSCs rapidly for five frames pre-bleach, then half of the GC was bleached, and recovery of NPM1 was monitored for two minutes.
(MP4)

**S3 Video. Cell division in control and BMH-21 treated hiPSCs.** Time-lapse imaging of control (left) and BMH-21 treated (right) hiPSCs undergoing cell division. The nucleolus

disassembles leading up to division with UBF remaining associated with rDNA. In control cells, the nucleolus reassembles following division whereas the nucleolus in BMH-21 treated cells fragments.

(MP4)

## Acknowledgments

The authors thank members of the Trainor lab for their discussions and insights into this work. We are also grateful to Marina Thexton and the Stowers institute for Medical Research Laboratory Animal Services core for animal care and husbandry, and to members of the Tissue Culture, and Microscopy Cores for their expertise and advice, and to the Virginia Commonwealth University Transgenic/Knockout Mouse Facility.

## Author Contributions

**Conceptualization:** Soma Dash, Maureen C. Lamb, Paul A. Trainor.

**Data curation:** Soma Dash, Maureen C. Lamb, Jeffrey J. Lange, Mary C. McKinney, Dai Tsuchiya, Fengli Guo, Xia Zhao, MaryEllen Kirkman, Emma L. Moore, Sean McKinney.

**Formal analysis:** Soma Dash, Maureen C. Lamb, Jeffrey J. Lange, Mary C. McKinney, Dai Tsuchiya, Fengli Guo, Xia Zhao, MaryEllen Kirkman, Emma L. Moore, Sean McKinney.

**Funding acquisition:** Soma Dash, Emma L. Moore, Rita Shiang, Paul A. Trainor.

**Investigation:** Soma Dash, Maureen C. Lamb, Jeffrey J. Lange, Dai Tsuchiya, Fengli Guo, Timothy J. Corbin, Emma L. Moore, Paul A. Trainor.

**Methodology:** Soma Dash, Maureen C. Lamb, Jeffrey J. Lange, Mary C. McKinney, Fengli Guo, Xia Zhao, Timothy J. Corbin, MaryEllen Kirkman, Kym Delventhal, Sean McKinney, Paul A. Trainor.

**Project administration:** Soma Dash, Maureen C. Lamb, Paul A. Trainor.

**Resources:** Timothy J. Corbin, Kym Delventhal, Rita Shiang, Paul A. Trainor.

**Supervision:** Soma Dash, Paul A. Trainor.

**Validation:** Soma Dash, Maureen C. Lamb, Jeffrey J. Lange, Dai Tsuchiya, Fengli Guo, Xia Zhao, MaryEllen Kirkman, Kym Delventhal, Emma L. Moore.

**Visualization:** Soma Dash, Maureen C. Lamb, Jeffrey J. Lange, Mary C. McKinney, Dai Tsuchiya, Fengli Guo, Xia Zhao, Timothy J. Corbin, MaryEllen Kirkman, Emma L. Moore, Sean McKinney.

**Writing – original draft:** Soma Dash, Maureen C. Lamb, Jeffrey J. Lange, Emma L. Moore, Paul A. Trainor.

**Writing – review & editing:** Soma Dash, Maureen C. Lamb, Jeffrey J. Lange, Mary C. McKinney, Dai Tsuchiya, Fengli Guo, Xia Zhao, Timothy J. Corbin, MaryEllen Kirkman, Kym Delventhal, Emma L. Moore, Sean McKinney, Rita Shiang, Paul A. Trainor.

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
