## [Decision Letter · Decision Letter 0]

7 May 2023

Dear Dr Trainor,

Thank you very much for submitting your Research Article entitled 'rRNA transcription is integral to phase separation and maintenance of nucleolar structure' to PLOS Genetics.

The manuscript was fully evaluated at the editorial level and by two independent peer reviewers, who differed in their enthusiasm for the work. Reviewer #1 had a number of serious concerns regarding interpretation of some of the cell biologic and molecular events, and recommended rejection; reviewer #2 had several minor concerns that could be addressed with additional discussion.

The manuscript and the reviews have now been discussed among members of the editorial board. While we appreciate and understand the concerns raised by reviewer #1, there is a consensus that the multiple mutant in vivo and hiPSC analyses represent an important contribution for the genetics community. Regarding specific comments from reviewer #1, we do not think direct measurement of rRNA synthesis or steady-state levels of Pol1 subunits (comment #4) arenecessary, nor do we think additional mutant studies, e.g. of PolR1F (comment #3), are needed. However, we do ask that you discuss the concerns raised by reviewer #1 regarding the relationship of nucleolar "remnants" described in your work to previous work on cell biology of the nucleolus (comments #1, #3). We also ask that you discuss additional concerns raised by reviewer #1 in confidential comments to the editor regarding the rigor of the conclusions regarding phase separation, and what additional studies might shed light on this question in the future.

We therefore ask you to modify the manuscript according to the review recommendations. Your revisions should address the specific points made by each reviewer.

Yours sincerely,

Gregory S. Barsh

Editor-in-Chief

PLOS Genetics

Gregory Copenhaver

Editor-in-Chief

PLOS Genetics

Reviewer's Responses to Questions

**Comments to the Authors:**

Reviewer #1: This manuscript demonstrates the requirement for rRNA transcription during development. Its particular value is the demonstration that the knockout of RNA polymerase I and III subunits results in nucleolar defects. While it is assumed that the lack of the polymerase subunits results in the inhibition of rDNA transcription and that the inhibition of embryo survival is due to the inhibition of rRNA synthesis. However, the authors do not measure rRNA synthesis. The Y10b antibody measures the steady state of the intermediates in the nucleolus and not their synthesis.

Other comments

1. It’s known that there are both prenucleolar bodies that contain components of the rDNA transcription apparatus as well as bodies that also contain rDNA. The lack of nucleolar RNA (lack of Y10b staining, Figure 1), would suggest that the nucleolar “remnants” observed may be reflective of protein-protein interactions that are independent of rRNA synthesis/processing. Might these structures be similar to those observed immediately post-mitosis prior to the resumption of rDNA transcription?

2.Pol R1C and PolR1D are both subunits of Pol I and Pol III. Hence, the inclusion of those mutants does not enhance the primary conclusions of the manuscript. What happens if you knockout PolR1F?

3. The measurement of the intensity of staining of a particular protein might be indicative of its relocation from the nucleolus as reported in Supplemental Figure 1. However, it may also reflect its destruction as well as the redistribution of the protein throughout the nucleolplasm/cytoplasm.

4.The authors miss the opportunity to determine if the knockout of a core Pol I subunit affects the steady-state levels of the other subunits as has been reported for the effects of BMH21.

Reviewer #2: This study by Dash et al. investigates the impact of ribosomal RNA (rRNA) transcription on nucleolar structure in mammalian development. Transcription of rRNA, catalyzed by RNA Polymerase I (RNA Pol I), is the essential first step in ribosome biogenesis. Loss of function variants in RNA Pol I subunits cause congenital conditions, called ribosomopathies, that disproportionately impact craniofacial development. Previously, the authors showed that craniofacial differences in ribosomopathies result from impaired rRNA transcription, activation the P53-mediated nucleolar stress response, and selective apoptosis in neural crest cells. Since nucleolar stress is closely linked with changes in nucleolar structure, this study aims to determine how loss of RNA Pol I alters nucleolar organization and assembly during development. Herein they present compelling evidence that loss of RNA Pol I activity, either through genetic inactivation in mouse embryos or pharmacological inhibition in human iPSCs, disrupts liquid-liquid phase separation that is necessary to establish the nucleolus’ tripartite structure and promote nucleolar reassembly following mitosis. In the absence of rRNA transcription, liquid-liquid phase separation is disrupted due to increased viscosity of the nucleolus’ outer granular compartment (GC). Overall, this is a comprehensive study that sheds new light on a requirement for rRNA transcription on nucleolar structure and function in development and, potentially, the pathogenesis of ribosomopathies.

General Feedback:

1. To investigate potential changes in the nucleolar phase separation properties following RNA Pol I inhibition, FRAP was used to follow recovery of RFP-tagged NPM1. It is not clear why phase separation was examined only using RFP-NPM1 (which marks the GC and not also GFP-FBL (which marks the dense fibrillar component (DFC)). rRNA transcription occurs at the FC–DFC interface where Pol I complexes are enriched. Higher concentration of rRNA decreases FBL saturation concentration to induce preferential condensation of fibrillarin in the DFC and promote phase separation. Therefore, changes in the nucleolar phase separation properties (increased viscosity) upon RNA Pol I inhibition may be, in part, due to increased FBL.

2. The models used here lead to complete loss of RNA Pol I activity (homozygous loss of function and pharmacological inhibition). There should be some discussion that addresses the following: How do these models shed light on ribosomopathies caused by partial loss of RNA Pol I activity? Is it expected that nucleolar morphology and assembly would be similarly impacted in cells where RNA Pol I activity is reduced but not completely lost?

3. How does this work reveal new insights into the phenotypic specificity for neural crest cells? For example, there is cell type-specific heterogeneity in the number of FC-DFC modules per nucleolus. Do the number of FC-DFC modules per nucleolus in neural crest cells make them more reactive to nucleolar stress in response to abnormal nucleolar structure?

Minor edits:

1. Define “NORs” for the reader.

2. Please clarify the following sentence: The whole FRAP recovery (Tau2, Fig 3O) also increased with BMH-21 treatment, suggesting there is reduced affinity, or increased resistance to the addition of NPM1 tothe GC underlying the increased recovery.

3. Why is the level of Fbl expression variably affected in the Polr1 mutants? (Figure 1)

4. That BMH-21 treatment of E8.5 embryos results in a single nucleolus and fragmented nucleoli phenotype could be more easily appreciated with higher magnification images of DAPI/Treacle/Pbl in Figure 3C and D.

**Have all data underlying the figures and results presented in the manuscript been provided?**

Reviewer #1: Yes

Reviewer #2: Yes

PLOS authors have the option to publish the peer review history of their article (what does this mean?). If published, this will include your full peer review and any attached files.

Reviewer #1: No

Reviewer #2: No

---

## [Decision Letter · Decision Letter 1]

3 Jul 2023

Dear Dr Trainor,

We are pleased to inform you that your manuscript entitled "rRNA transcription is integral to phase separation and maintenance of nucleolar structure" has been editorially accepted for publication in PLOS Genetics. Congratulations!

The revised manuscript has been seen by the two prior reviewers, both of whom (as you will see from their comments below) recommend moving forward.

Yours sincerely,

Gregory S. Barsh

Editor-in-Chief

PLOS Genetics

Gregory Copenhaver

Editor-in-Chief

PLOS Genetics

Comments from the reviewers (if applicable):

Reviewer's Responses to Questions

**Comments to the Authors:**

Reviewer #1: The authors have addressed the questions this reviewer raised. While I believe that further experiments would go a long ways to proving their central point, the results contained in the manuscript provide are interesting and should provoke further experimentation.

Reviewer #2: The feedback from the prior review has been addressed.

**Have all data underlying the figures and results presented in the manuscript been provided?**

Reviewer #1: Yes

Reviewer #2: Yes

PLOS authors have the option to publish the peer review history of their article (what does this mean?). If published, this will include your full peer review and any attached files.

Reviewer #1: No

Reviewer #2: No

**Data Deposition**

http://datadryad.org/submit?journalID=pgenetics&manu=PGENETICS-D-23-00356R1

**Press Queries**

---

## [Editor Report · Acceptance letter]

24 Aug 2023

PGENETICS-D-23-00356R1 

rRNA transcription is integral to phase separation and maintenance of nucleolar structure 

Dear Dr Trainor, 

We are pleased to inform you that your manuscript entitled "rRNA transcription is integral to phase separation and maintenance of nucleolar structure" has been formally accepted for publication in PLOS Genetics! Your manuscript is now with our production department and you will be notified of the publication date in due course.

With kind regards,

Zsofia Freund

PLOS Genetics

On behalf of:
